# Suicide Prevention for International Students: A Scoping Review

**DOI:** 10.3390/ijerph20021500

**Published:** 2023-01-13

**Authors:** Samuel McKay, Maria Veresova, Eleanor Bailey, Michelle Lamblin, Jo Robinson

**Affiliations:** 1Orygen, Parkville, VIC 3052, Australia; 2Centre for Youth Mental Health, The University of Melbourne, Parkville, VIC 3010, Australia

**Keywords:** suicide prevention, international students, foreign students, education settings

## Abstract

International students are at risk of suicide and suicide prevention activities addressing their unique needs are required. However, no comprehensive review has been undertaken to identify effective suicide prevention approaches for international students. The current scoping review aimed to chart the extent, range, and nature of available evidence on the prevention strategies for international students. We systematically searched Medline, PsycInfo, ERIC, CINAHL, Proquest, and several gray literature databases to identify relevant peer-reviewed articles and gray literature. Eligible publications were those providing data or recommendations related to suicide prevention among international students; 15 peer-reviewed articles and 19 gray literature documents were included in the review. No studies of prevention programs or policies directly targeting suicidal ideation, suicide attempts, or suicide in international students were identified. A narrative synthesis of the suicide prevention recommendations for international students identified four categories: (1) cultural competency training on suicide and provision of culturally sensitive services; (2) improved and increased risk screening for suicide; (3) proactive intervention and engagement strategies; and (4) collaborative approaches to streamline service access and improve available support. These recommendations provide guidance on potential directions for international student suicide prevention. Research assessing the effectiveness of such recommendations will enable the development of novel evidence-based policies and interventions that reduce rates of suicide in international students.

## 1. Introduction

Suicide is a leading cause of death for young people, and students studying at post-secondary education institutions are a key group at risk of suicide-related thoughts and behaviors, including suicidal ideation, suicide attempts, and death by suicide [1,2]. Suicide prevention programs may reduce suicide risk in higher education students [2,3], and education settings are well placed to deliver suicide prevention activities [2,4]. However, some groups of students in higher education settings may require specific suicide prevention programs that are designed to meet their unique needs [5].

International students who complete their studies in a different country to their primary place of residence comprise a significant portion of the tertiary education student body in many countries [6] and, like many migrant groups, experience a distinct set of stressors [7] and barriers to help seeking that can increase their risk of suicide [8,9]. Key stressors for international students include language difficulties, processes of acculturation, experiences of discrimination and racism, social isolation, financial issues, and academic pressure [7]. The combination of these factors can take a toll on students’ mental health [7], increase suicidal ideation [10], and have been implicated in suicide deaths of international students [11]. Unfortunately, international students can also face significant barriers to help seeking and service engagement due to cultural stigma related to mental health and lower mental health literacy [8]. It has been suggested that prevention strategies and programs for international students need to address this unique combination of stressors and barriers to service engagement in order to maximize effectiveness [8,12]. However, no synthesis of the literature is currently available that provides a comprehensive overview of the risk and protective factors or prevention strategies for suicide among international students [13].

There were three main objectives of the current review. The first was to chart the extent, range, and nature of available evidence on suicide prevention strategies in the international student community. The second was to identify gaps and limitations in the literature and provide future research recommendations to address them. The third was to provide guidance for suicide prevention policies and best practice guidelines for government, the education sector, health service providers, and other groups that service international students.

## 2. Materials and Methods

### 2.1. Overview

The current review utilized the enhanced framework of scoping reviews [14], which is based on the methodology of Arksey and O’Malley [15]. The framework involves six stages: (1) identifying the research question; (2) identifying relevant studies; (3) selecting studies; (4) mapping/charting the data; (5) collating, summarizing, and reporting the results; and (6) expert consultations. A full explanation of each of these steps is available in our review protocol [13] and a brief summary is provided here.

### 2.2. Step 1: Identifying the Research Question

Research questions for scoping reviews need to have enough breadth to capture the range of literature on a topic, while also clearly outlining the scope of the review [14]. The current review utilized the population (e.g., international students), concept (e.g., prevention strategies), outcome (e.g., suicide, suicide attempts, and suicidal ideation), context (e.g., postsecondary education settings worldwide) model from the PRISMA-ScR guidelines [16] when developing the following research questions:What is the extent, range, and nature of the evidence regarding suicide prevention for international students?What suicide prevention strategies are promising for preventing suicide in international students?

### 2.3. Step 2: Identifying Relevant Studies

A comprehensive systematic search strategy was developed with the support of a librarian that aligned with the guidance from the Joanna Briggs Institute Reviewer Manual for scoping reviews [17]. Five academic databases were searched: Medline (EBSCOhost), PsycInfo (Ovid), ERIC (EBSCOhost), CINAHL (EBSCOhost), and ProQuest Dissertations and Theses. A systematic approach was also taken to search the gray literature. The first step involved searching the following gray literature databases: Google Scholar, Open Grey, Trove—The National Library of Australia, the British Library, ResearchGate, Science.gov, and UNESCO. To identify relevant governmental and organizational reports and policy documents, country-specific searches were conducted on Google for the United States of America (USA), Canada, the United Kingdom, and Australia, which are the top four English-speaking destinations for international students. We also searched Open Science Framework (OSF) preprints and Proquest to identify relevant preprint and theses documents. The two sets of keywords used for all gray literature searches were “international student suicide” and “international student suicide prevention”. Only the first 100 hits in the gray literature databases were screened, as further screening was unlikely to yield additional relevant literature [18]. Finally, we contacted subject matter experts to identify any evidence that was not identified through the initial searches.

### 2.4. Step 3: Study Selection

Three members of the research team (SM, MV, and EB) independently conducted the screening of the titles and abstracts, as well as the full text screening. Any disagreements were discussed by the full screening team until a consensus was reached. Questions raised during the screening process and associated decisions can be found in Appendix A. Literature was excluded if it: (1) did not specifically address international students; (2) did not address prevention strategies or provide recommendations for suicide prevention; (3) did not address suicidal ideation, attempts, or death by suicide as outcomes; or (4) was not focused on postsecondary education settings.

### 2.5. Step 4: Charting the Data

Data extraction was conducted by two reviewers (SM and MV) with a first reviewer identifying relevant information and the second reviewing the initial extracted information for errors or omissions. Data extraction of the peer-reviewed studies was undertaken in Covidence [19] using predefined data that captured study characteristics (year, country), aim, design, setting, population (e.g., age, gender, ethnicity/country of origin), outcome (e.g., prevalence, suicide, suicidal ideation, tested prevention strategy, presence or absence of a control group), any suicide risk or protective factors (e.g., correlates or predictors of suicide, suicide attempt, or suicidal ideation), finding interpretations, recommendations for future research, and policy. Methodological quality of the peer-reviewed literature was assessed using the Standard Quality Assessment Criteria for Evaluating Primary Research Papers tool [20]. Scores were converted to a percentage to create a consistent comparison metric; with >80% considered strong, 70–80% considered good, 55–69% considered adequate, and <55% considered limited [20].

The gray literature was extracted using a predefined template in Excel that captured the author, document type, country, year, a summary of any empirical findings, and recommendations for future research and policy.

### 2.6. Step 5: Collating, Summarizing, and Reporting the Results

The data from the selected literature were collated based on the PRISMA-ScR checklist guidelines for reporting results [16]. Quantitative results reported on the study characteristics and recommendations for the peer-reviewed (see Table 1) and gray literature (see Table 2). A narrative synthesis of the suicide prevention recommendations was conducted using a thematic approach (see Table 3). Raw data were coded and thematically clustered to identify patterns and overarching categories. These patterns and categories were reviewed and revised by the study team and relevant experts in an iterative process.

### 2.7. Step 6: Consultation

Two provider partners and three subject matter experts reviewed and provided feedback on the results of the review to ensure that our interpretations were applicable, understandable, and included all relevant documents and information.

## 3. Results

There were 74 records retrieved through our search of the peer-reviewed literature and 14 duplicates were removed using Covidence’s duplicate removal function after all potential duplicates identified by the system were manually reviewed. After duplicate removal, 60 records were included for title and abstract screening; 29 passed the initial screening and were reviewed as full text papers, with 15 studies included in the final review. See the PRISMA flow diagram in Figure 1 for a full summary of the academic database search process. Table 1 summarizes the key study characteristics of the included academic articles.

The gray literature search yielded 817 potentially relevant records, with 616 to review after duplicates were manually removed. A total of 19 of those records progressed through to the extraction phase after full text review. Table 2 presents a detailed overview of the included gray literature documents.

### 3.1. Peer-Reviewed Study Characteristics

The majority of peer-reviewed published literature was based on international student samples studying in the USA [1,12,21,22,24,26,27,28,29,30,31], with three studies conducted in Australia [8,23,25] and one in Japan [10]. Most studies were cross-sectional [8,10,12,21,25,27,28,29,31], with a small number of case studies [24,26], one case series [1], a qualitative study [23], a literature review, and an opinion piece [30]. There were no longitudinal studies identified. All studies that included international student samples addressed risk and protective factors related to suicidal ideation or suicide-related behaviors. Sample sizes ranged from 9 to 2423 (M = 435.20, SD = 726.44), with a total of 4352 participants. There was an overall higher proportion of females (58.6%, range = 48.4–80.9%) than males across the samples. When the country of origin of the international student sample was reported, most participants were from Asian countries and all included samples were based in university settings. Study quality for publications that included international student samples was, overall, rated as high (M = 82.4%, range = 65.0–95.5%). Three studies not involving international student samples directly reported on the development of policies or services focused on suicide prevention or postvention among international students at universities [22,24,26]. All included studies provided at least one recommendation for future suicide prevention activities.

### 3.2. Gray Literature Characteristics

Similar to the peer-reviewed articles, most gray literature documents were from the USA [35,37,38,39,40,47,48,49], followed by Australia [11,32,33,34,36,41,46] and the United Kingdom [42,43]. There was a large variety of document types, including theses, reports, government submissions, and best practice guides. Four of the gray literature documents included empirical data related to suicide in international students. Two of these studies assessed international student suicides in Australia between 2009 and 2019, identifying key risk factors associated with the students’ deaths [11,36]. The other two studies explored risk factors associated with suicidal ideation, suicide attempts, and suicide deaths in international students in the USA [38,49]. All included gray literature provided a minimum of one recommendation for suicide prevention activities.

### 3.3. Intervention Studies

No studies assessing specific suicide prevention interventions were identified in either the peer-reviewed or gray literature. Given this, we adjusted our focus to assess published recommendations for suicide prevention with international students to identify current best practice recommendations and future research opportunities.

### 3.4. International Student Suicide Prevention Recommendations

Table 3 shows the summary of suicide prevention recommendations identified in the peer-reviewed and gray literature. Four main themes of recommendations were identified related to suicide prevention for international students. The first focused on improving service delivery for international students through cultural competency training for key stakeholders. This training would address the key risk factors for suicide and barriers to help seeking, along with adapting counseling services to international student needs. The second centered on improved and increased risk screening for suicide by mental health services (e.g., formal risk screening using culturally appropriate tools) and within the community (e.g., gatekeeper models) for international students. The third theme addressed proactive intervention and engagement strategies targeting the key risk factors for suicide and barriers to help seeking. A key component of this approach is the utilization of peer-based programs. Finally, the fourth theme covered collaborative approaches between key service providers that could be used to streamline service provision across university and mental health settings. There was an emphasis on the need to include international students and those who work with them in the development and implementation of any service provision changes.

An overarching aspect of many of the recommendations was related to policies and procedures. In particular, it was noted that institutions should have a suicide protocol that included the international student office (e.g., the support services for international students supplied by an education provider) both in terms of development and implementation [38]. It was further recommended that the protocol specifically include crisis and postvention planning (e.g., an organized response after a completed, attempted, or suspected suicide occurs) that engages the international student office (we did not include gray literature that only recommended that an international student office at a specific university should be involved in the case of an international student death because it was not relevant to the broader recommendations; however, it is worth noting that this was a common feature of the excluded gray literature documents.) [34], along with provisions for engaging embassy and interpreters when interacting with international family members [50]. Finally, broader policy recommendations focused on implementing a specific reporting framework for international student suicides to better track the scope of the problem [33,36].

### 3.5. Future Research Recommendations

Future research recommendations were provided in 12 of the 15 peer-reviewed articles. These covered potential topics along with methodological and conceptual issues that could be addressed in future studies. In terms of research topics, more studies are needed that assess interventions based on identified risk factors and barriers to evaluate effective approaches for suicide prevention (e.g., gatekeeper training), replicate existing studies with international students from countries other than Asia and more diverse cohorts [25,28,29,31], and assess a broader range of risk and protective factors at the individual level (e.g., coping styles) and macro level (e.g., host receptivity and the size of existing ethnic communities on campus) [12,51]. As for methodological and conceptual issues, the existing studies suggested a need for longitudinal and experimental designs to assess the temporality and causality of effects [12,25,27,28,29], alternative measures of the risk and protective factors that have been previously studied [29], and qualitative or mixed methods studies that could provide richer information about the mental health experiences of international students [8,12,25,28,29,31]. Additionally, it was recommended that future research assesses differences between international student cultural groups [31] and separates out the impact of pre-existing mental health issues with those that develop or are exacerbated by studying in another country [27]. Finally, there was a call to use more advanced statistical modeling techniques in future studies [10].

## 4. Discussion

The overarching aim of this review was to provide a comprehensive synthesis of the available literature on suicide prevention with international students that could be used to guide future prevention policy and actions. This scoping review included 15 peer-reviewed articles and 22 gray literature documents. The discussion overviews the key insights derived from the review and contextualizes the findings within the larger existing evidence and policy landscape. Recommendations for future research and policy are provided.

### 4.1. The State of the Literature

#### 4.1.1. Characteristics of Existing Evidence

The majority of available literature on international student suicide has used correlational or other similar designs that do not allow for conclusions on the causality of effects. Similarly, most studies have relied on samples from a small number of countries or single sites, and there has been an overrepresentation of Asian international students, limiting conclusions that can be drawn regarding other countries, sites, or groups. Sample sizes in many studies were also small, reducing the power to identify small significant effects or generalize the findings. In terms of outcomes, most of the research has focused on suicidal ideation, with only a small number of studies assessing suicide attempts or deaths. Finally, some studies drew upon existing theories of suicide, with most of these utilizing Joiner’s interpersonal theory of suicide and the key associated variables of perceived burdensomeness and belonging [27,28,29].

#### 4.1.2. Suicide Risk and Protective Factors

The literature in this review has identified several key risk and protective factors for suicide and suicidal ideation in international students. Male and younger (18–24 years) international students have been shown to be at the greatest risk of dying by suicide [11,36], which aligns more generally with the youth suicide literature [52]. This knowledge may be helpful for practitioners or others who work with international students, so that they can screen for such risk. Although, it is important to note that the clinical utility of available risk screening measures for predicting suicidal behavior is currently very limited [53]. Indeed, recent advice from the National Institute for Health and Care Excellence (NICE) in the UK recommends against using such tools to stratify suicide risk levels, predict future suicides, or determine who receives treatment [54]. Instead, when potential suicide risk is identified via screening measures or gatekeepers (e.g., community members trained to identify risk), a collaborative effort focused on detecting and addressing modifiable risk factors is recommended [53,54]. Such modifiable risk factors can also be used to guide broader prevention efforts for international students [55].

One of the most commonly identified modifiable risk factors was academic stress, which can be heightened by perceived discrepancy with family or the students’ own performance expectations [29], or when a student fails a course [36]. Financial stress was found to be present when international students died by suicide [11]. Experiences of isolation, loneliness, racism, discrimination, and perceived burdensomeness have also been found to heighten suicidal ideation [10,12,23,25,28]. In contrast, a sense of campus belongingness and a general sense of connectedness, along with problem-focused coping, can act as protective factors against suicidal ideation [10,25,28].

As for barriers to help seeking, international students tend to have lower mental health literacy and help seeking intentions for suicidal ideation than their domestic peers [8]. They are less likely to engage with services, have a mental health diagnosis, or continue using services when they do engage [11,31,38,49]. The evidence suggests that this may be because available services are not well adapted to international students’ needs [11,37,44].

#### 4.1.3. Intervention Studies

The biggest gap identified in the literature is the lack of suicide prevention interventions designed for, or evaluated with, international students. This limits any conclusions on effective intervention strategies to reduce suicide risk in this population. However, there is an extensive range of recommendations aimed at reducing international student suicide available in both the academic and gray literature that align with the broader literature on suicide prevention with individuals from culturally and linguistically diverse (CALD) communities [56]. These provide a good foundation for potential future intervention studies, and some guidance on how to minimize risk of suicide in international students. Similarly, there are a range of existing suicide prevention programs that have been shown to be effective in education settings more broadly [2,4]. For instance, universal gatekeeper training has been shown to increase skills related to the identification of suicide risk, and targeted interventions for at risk students have been found to reduce suicidal ideation in domestic students [2]. These interventions could be adapted to be appropriate and acceptable for international students. Nevertheless, until the existing recommendations or any existing interventions are adapted and evaluated with international students, we cannot be sure of their effectiveness in reducing the risk of suicide for this group.

### 4.2. Research and Policy Directions

As this review highlights, the field of suicide prevention for international students is small but growing, and the existing literature points to several directions for future research and policy. In terms of research, there is a need to develop and test potential interventions to identify which approaches are most effective across different international student groups and countries [25,28,29,31]. Ideally, these studies should employ a randomized controlled design. However, given the lack of existing evidence, pre-post studies could be used to initially determine potentially effective interventions. Similarly, further studies assessing potential risk and protective factors across different groups using qualitative or mixed method designs would facilitate opportunities to identify novel factors that are missing in the current literature [8,12,25,28,29,31,51]. Future research should focus on the development of measures that are appropriate for different international student groups and can be used to study suicide-related behaviors and the associated risk and protective factors more effectively in these cohorts. A focus on involving international students with a lived experience of suicide-related thoughts and behaviors should be a priority, as such input can support the feasibility, acceptability, and cultural sensitivity of any research findings or prevention programs [28,45,46,49].

Opportunities also exist to draw upon a wider range of theories and models from the suicide, acculturation, and public health literature when developing and testing potential interventions or prevention programs for international students. For example, comprehensive suicide prevention strategies often draw upon a public health framework that stratifies prevention into universal (i.e., delivered to the whole population), selective (i.e., delivered to groups at higher risk of suicide), and indicated (i.e., delivered to people showing suicide-related behaviors) approaches [57]. Combined approaches addressing each of the levels of this framework have been shown to be effective with young people [57], and this model could be used to develop comprehensive suicide prevention interventions for international students.

In terms of both policy and research, the current review identifies several key areas for suicide prevention among international students relevant to educational institutions, government, mental health service providers, and other organizations who work with international students. In particular, suicide prevention activities and programs should include: culturally sensitive service provision (e.g., staff training, culturally appropriate service settings) [22,27,30,37,38,47,49], risk screening (e.g., formal screening, gatekeeper training, addressing modifiable risk factors, etc.) [1,40], proactive engagement strategies (e.g., information campaigns, mental health training, peer-based programs, etc.) [10,12,25,30,35,37], and streamlined services across settings (e.g., university, mental health providers, hospitals, etc.) [12,24,28,31,35,36,45,46,49]. The development of interventions should involve international students and those who closely work with them [28].

Policymakers may find that innovative approaches to this problem require a combination of the different intervention approaches to be effective. Importantly, structural issues related to equity and access to care should also be a focus of governments and other policy-makers who have the power to make the necessary systemic changes to provide appropriate accessible care for international students [8,11]. Finally, there is an increasing awareness of the need for clear postvention plans in education settings. Where available, national bodies can provide the required guidance on such plans. For instance, general national postvention guidelines for university settings have been developed in Australia, the UK, and the USA [58,59,60]. However, a greater focus on international student needs could increase the applicability of such guidelines for this group in the future.

### 4.3. Limitations

This study has several limitations. First, we set out to review studies specifically assessing suicide prevention interventions with international students, but were only able to report on prevention recommendations due to the lack of intervention studies. Second, we only included studies in English and focused on a small number of key destination countries for international students as part of our gray literature search. This means we may have missed important findings in other languages and from other places. Third, the recommendations for suicide prevention in this review are based on a wide variety of sources, limiting our capacity to be sure of the quality of each of the recommendations. Finally, although the quality appraisal tool that we used to assess the peer-reviewed research quality in this study was well adapted to the diversity of research in the field, it provides a somewhat optimistic view of the quality of the literature.

## 5. Conclusions

International students are a group who experience a number of factors that place them at an increased risk of suicide. This review provides a comprehensive overview of the current state of the suicide prevention literature and associated recommendations for international students. Significantly, this review found that no suicide-specific interventions for international students currently exist or have been evaluated. The existing suicide prevention recommendations point towards numerous gaps in current systems that can be addressed to better support international students and reduce their risk of suicide. Studies assessing the efficacy of these different approaches should be a priority so we can develop an evidence base of effective methods that reduce suicide risk in international students. Similarly, research identifying novel or innovative approaches to suicide prevention with international students would also be valuable for guiding future prevention efforts.

## Figures and Tables

**Figure 1 ijerph-20-01500-f001:**
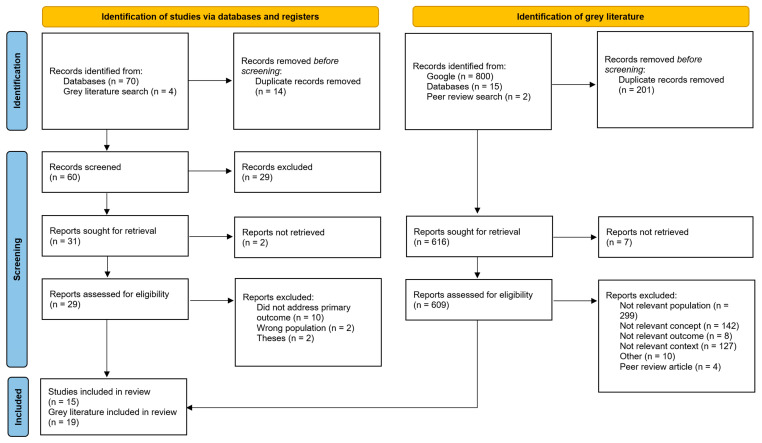
Preferred Reporting Items for Systematic Reviews and Meta-Analyses flow diagram. Note: Other = Includes documents such as presentation slides that were considered ineligible for this review (see Appendix A for further information).

**Table 1 ijerph-20-01500-t001:** Peer-Reviewed Studies on Suicide Prevention among International Students.

Author(s)	Year	Country	Study Design	Study Aim	Sample	Main Findings for International Students	Suicide Prevention Recommendations	Future Research Recommendations	Quality Appraisal
Cerel et al. [21]	2013	USA	Cross-sectional study	To determine experiences regarding suicide and those affected by it, and to determine students’ attitudes, perceptions, and behavioral intentions about suicide prevention resources	117 college (9.3% international) students. M_age_ = 23.79, SD = 6.88. 36.8% male, 62.4% female, 0.8% unidentified. Demographics for international students not separately provided. Country of origin not specified.	International students less likely than domestic students to be aware of Lifeline (telephone support line for suicide) or see suicide as a problem	Campus resources should be directed towards international students to better engage them in services	Studies assessing effectiveness of targeted messaging for specific student groups, such as international students	77.3%
Choi et al. [22]	2020	USA	Text and opinion	N/A	No participants. Editorial on Asian international students from the perspective of MDs treating this population	N/A	Medical practitioners require knowledge of acculturation processes Development of international-student-specific risk screening tools Medical practitioners should assess patient personality, utilize appropriate treatment modalities, and solution-focused approaches for suicidal ideation	Studies delineating impairment and suicide risk in international students	N/A
Clough et al. [8]	2019	Australia	Cross-sectional study	To examine potential differences in mental health and related constructs, such as mental health literacy and help-seeking attitudes, between domestic and international students in Australia	209 international university students. M_age_ = 23.02, SD = 5.41. 37.5% male, 62.5% female. Country of origin not specified. 148 Domestic university students. M_age_ = 25.34, SD = 9.32. 19.1% male, 80.9% female	International students had lower mental health literacy, help seeking attitudes, and help seeking intentions for suicidal ideation than domestic students	Proactive interventions targeting mental health and help seeking for international students	Mixed methods or qualitative designs to facilitate insight into specific target areas for future mental health interventions for international students	81.2%
Dovchin [23]	2020	Australia	Qualitative research	To illustrate how international students in Australia experience linguistic racism through ethnic accent bullying and linguistic stereotyping, and how the combination of this overall harmful experience may further cause psychological inferiority complexes, leading to mental health issues such as depression, suicidal ideation, and social anxiety disorder	Nine international university students. M_age_ = 23.55, SD = 3.78. 33.3% male, 66.7% female. Two participants from China, two from Mongolia, and one each from Vietnam, Somalia, Ukraine, Singapore, and Hong Kong	Experiences of linguistic racism (e.g., negative attitudes towards spoken accent or language pronunciation) reported to increase suicidal ideation by participants	Linguistic racism and associated bullying should be the target of suicide prevention programs Training of university personnel including mental health professionals on impacts of linguistic racism and bullying required	None provided	65.0%
Hong et al. [1]	2022	USA	Case series	To examine variation in clinical characteristics, including suicidal ideation, suicide attempt history, and non-suicidal self-injury, across socio-demographic subgroups of students presenting for psychiatric emergency services	725 college students (8% international students) visiting psychiatric emergency services. M_age_ = 22.00, SD = 3.97, 43.9% male, 56.7% female. Country of origin of international students not specified	International students were more likely to report lifetime history of multiple suicide attempts than domestic students International students were involuntarily admitted to the hospital a significantly higher percentage of the time than domestic students	Proactive screening and outreach efforts needed to reach international students and link them to services before crises emerge.	Studies of effective interventions are needed	86.4%
Kaslow et al. [24]	2012	USA	Case study	To provide an example of an effective suicide prevention coalition	University staff and coalition members (unspecified number and no demographic information provided) who participated in the development of an effective campus wide coalition for suicide prevention	Effective coalitions targeting international students recruit and involve diverse community members in committees, including international students and those who have expertise in international student needs (e.g., international office, faculty from diverse backgrounds, international colleagues, and other relevant stakeholders) Development of language-specific communication materials (e.g., website and videos) for international students by international students supported creation of engaging materials for international student cohort	Include international students and those who work with them in development of suicide prevention activities, outreach programs, and any broader campus-based suicide prevention coalition activities	None provided	N/A
Low et al. [25]	2022	Australia	Cross-sectional study	To assess whether loneliness, campus connection, and problem-focused coping moderate the relationship between stressful life events and suicide ideation in Asian international students studying in Australia	138 Asian international students M_age_ = 21.00, SD = 1.86, 32% male, 68% female. The origin country of participants was Singapore (44.2%), Malaysia (39.9%), Indonesia (8.7%), Hong Kong (3.6%), India (1.4%), Sri Lanka (0.7%) and Thailand (0.7%)	The relationship between stressful life events and suicidal ideation was moderated by lower levels of loneliness, higher levels of campus connectedness, and problem-focused coping	Suicide prevention efforts should address the issues of loneliness and lack of campus connectedness among international students. Activities aimed at promoting student cohesiveness on campus and peer support groups (e.g., camping trips, coffee clubs, or movie nights) at universities can be used to address such issues Increase awareness and skills related to problem-focused coping in international students. University counseling clinics could host workshops or promote such skills as part of their services	Qualitative data are required for future research to provide a rich and more accurate perspective of the experience of Asian international students living in Australia Further research needs to examine which of the key variables of campus connectedness, loneliness, or problem-focused coping are the most important for reducing the impact of stressful life events on suicidal ideation and how such relationships longitudinally impact suicidal ideation	90.1%
Meilman and Hall [26]	2006	USA	Case study	To describe the development and successful implementation of campus postvention services in the aftermath of college student deaths by suicide, as well as by natural and accidental causes	Two university staff responsible for development of postvention approach at a single university	Postvention Community Support Meetings support students and staff to process tragedy of student deaths	Include international office in postvention planning and incident response	None provided	N/A
Nguyen et al. [10]	2021	Japan	Cross-sectional study	To explain how suicidal thoughts arise and persist inside one’s mind using a multifiltering information mechanism called Mindsponge	268 university students (75.0% international, 25.0% domestic). M_age_ = 20.87, SD = not reported. 36.6% male, 63.4% female. The origin regions of the sample were Japan (25.8%), Southeast Asia (45.5%), East Asia (17.9%), South Asia (6.7%), and Other (4.1%)	Sense of connectedness and companionship reduces suicidal ideation less in international students than in domestic students Sense of connectedness and companionship increases informal help seeking, but this effect is reduced in international students Sense of connectedness and companionship decreases formal help seeking, and this effect is more pronounced in international students	Interventions should take a systematic and coordinated approach targeting different risk factors, including sense of belongingness, accessibility of help-seeking sources, and reducing improper cultural responses to mental health issues (e.g., stigma)	More studies using Mindsponge modeling techniques	77.2%
Pérez-Rojas et al. [27]	2021	USA	Cross-sectional study	To examine the relationships among discrimination, cross-cultural loss, academic distress, thwarted belongingness, perceived burdensomeness, and suicidal ideation in international students	595 international college students from two universities. M_age_ = 24.57, SD = 4.56. 51.5% male, 48.0% female, 0.5% other. The origin countries of the sample were India (34.8%), China (19.1%), Vietnam (3.4%), Malaysia (2.4%), Colombia (2.2%), Indonesia (2.0%), South Korea (2.0%), with the remainder coming from 67 other countries worldwide	Perceived burdensomeness predicted increased suicidal ideation Discrimination, cross-cultural loss, and academic distress positively predicted sense of burdensomeness, but only academic distress directly contributed to suicidal ideation when controlling for burdensomeness Thwarted belonginess did not predict suicidal ideation	Messaging from campus staff and university should be assessed to minimize risk of increasing sense of burdensomeness in international students Cultural competence and suicide prevention training should be provided to academic advisors so they can identify risks and refer international students to service Cultural competence training should be provided to clinicians to help recognize discrimination, cross-cultural loss, and academic distress as risk factors for suicide in international students	Longitudinal research to test temporality of effects with broader populations (e.g., international students from non-Asian countries) and measures (e.g., university specific belongness) Studies that separate out impact of pre-existing mental health issues compared to new or exacerbated issues related to moving to a new country	90.1%
Servaty-Seib et al. [28]	2015	USA	Cross-sectional study	To assess the relationships between two types of belongingness (i.e., campus, family), perceived burdensomeness, and suicidal ideation in domestic and international students	254 college students (46 international, 208 domestic). M_age_ = 21.1, SD = 1.8. 51.6% male, 48.4% female. The country of origin of the international students in the sample was China (8.2% of sample), India (3.5% of sample), and Malaysia (1.6% of sample), and Kenya (1.0% of sample)	Perceived burdensomeness predicted increased suicidal ideation in international and domestic students High family belonginess related to greater suicidal ideation in international students High campus belonginess related to lower suicidal ideation in international students Perceived discrimination not related to suicidal ideation	Clinicians should encourage international students to find ways to enhance and maintain connection with campus Collaboration between clinicians, international student office, and other support groups could be used to enhance students’ sense of connection with their campus Campus-based needs assessment could be used to guide actions to increase international student sense of belonging at their campus Training could be created to engage domestic students to better include international students and enhance campus connection	Studies assessing similar models with international students from other countries and universities Longitudinal studies to test temporality of effects over time and in different school years Qualitative studies assessing cultural differences across groups in perceptions of thwarted belongingness and burdensomeness	81.2%
Taliaferro et al. [12]	2020	USA	Cross-sectional study	To assess the risk and protective factors that are associated with emotional distress and suicide ideation (i.e., strength and frequency of thoughts about killing one-self) among international college students	334 international college students. Age range 18–26 years, 56% female. Race/ethnicity of the sample was Asian (62.0%), European (11.0%), Hispanic/Latino (9.3%), African (4.8%), Middle Eastern (4.8%), and Other (7.8%)	Higher entrapment, unmet interpersonal needs (thwarted belonginess and perceived burdensomeness), and ethnic discrimination associated with increased emotional distress Cultural stress, family conflict, perfectionism, ethnic discrimination, and unmet interpersonal needs positively related to suicidal ideation, but only unmet interpersonal needs related to greater suicidal ideation when controlling for other variables	Clinicians should consider entrapment, ethnic discrimination, and unmet interpersonal needs as risk factors for suicide Increase sense of campus belonging for international students Decrease sense of burdensomeness of international students by emphasizing value they bring Develop campaigns to reduce systematic and individual ethnic discrimination Cooperation between key university departments (e.g., international office, counselling center, academic affairs, etc.) to produce programming that supports students to feel a sense of belonging on campus	Longitudinal studies to assess temporality of effects that include additional variables such as coping strategies Qualitative studies to understand international student experiences and identify possible prevention and intervention approaches Assessment of macro-level factors such as cultural and institutional patterns, including host receptivity, pressure to conform, and the size of existing ethnic communities on campus	77.3%
Wang et al. [29]	2013	USA	Cross-sectional study	To examine the moderating effects of three risk factors: perfectionistic personal discrepancy, perfectionistic family discrepancy, and discrimination on the associations between interpersonal risk factors (i.e., perceived burdensomeness and thwarted belongingness) and suicide ideation	Asian international university students (N = 466). M_age_ = 26.39, SD = 4.99. 50.43% male, 49.57% female. The sample ethnic subgroups included China (52.8%), India (14.8%), Korea (8.4%), Vietnam (6.4%), Taiwan (4.5%), Thailand (2.8%), Sri Lanka (2.1%), Indonesia (1.5%), Japan (1.5%), Malaysia (1.1%), Nepal (1.1%), Philippines (0.9%), and five other smaller subgroups	Maladaptive perfectionism in the form of personal and family discrepancy (failing to meet standards or expectations) and discrimination positively related to suicidal ideation Family discrepancy and perceived discrimination intensified relationship between perceived burdensomeness and thwarted belonginess and suicidal ideation.	Clinicians should assess discrepancy between Asian international students’ family expectations and actual academic performance, and impact on suicidality Provide effective coping methods for any perceived discrepancy Those working with Asian international students should normalize stress and negative impact of cross-cultural transition process on academic performance, as maladaptive perfectionism may be heightened when transitioning into a new academic context Clinicians should help Asian international students understand racial dynamics in host country and externalize discrimination and self-blame	Studies assessing similar models in more diverse international student cohorts Longitudinal experimental research that facilitates greater causal understanding of effects Alternative measures of discrepancy (e.g., parent or sibling report) and ethnic discrimination (e.g., language discrimination) could facilitate a more nuanced understanding of impact of these variables on suicidality Qualitative studies could provide richer information on influence perfectionism and discrimination on suicide risk	95.5%
Wong [30]	2013	USA and Canada	Text and opinion	N/A	No participants, review chapter on risk factors for suicide in diverse college students including international students	N/A	Culturally sensitive and community-based evidence-based interventions should be developed Cultural competency training should be provided to mental health professionals	Studies to determine efficacy of various approaches to suicide prevention	N/A
Yeung et al. [31]	2021	USA	Cross-sectional study	To assess the prevalence and correlates of mental health symptoms and diagnoses in international college students in the United States	44,851 university students (2423 international, 42,428 domestic) International: 37.8% male, 59.3% female, 2.5% other. Country of origin not reported Domestic: 29.3% male, 67.8% female, 2.5% other	International students less likely than domestic students to report mental health diagnosis International students more likely than domestic students to report suicide attempts and feeling overwhelmingly depressed.	Increase awareness and improve psychoeducation on reducing stigma of mental health problems and understanding of warning signs for suicides in culturally sensitive manner Peer-based mental health awareness and referral training to reduce stigma and increase service engagement Engage parents in mental health to further support international students Translate psychoeducation materials and recruit more linguistically diverse clinicians to support students who prefer to interact in their own language	Qualitative research into subjective mental health experiences of international students Further research into mental health of non-Asian international students Mixed methods research to explore similarities and differences across cultural groups that can facilitate more nuanced approaches to addressing mental health issues	81.8%

Note: Not all percentages add to exactly 100% due to rounding. Where possible means and standard deviations (SD) were reported to two decimal places, but in some cases, not all data was reported and when this occurred the available data were reported. N/A = Not applicable.

**Table 2 ijerph-20-01500-t002:** Gray literature on Suicide Prevention among International Students.

Authors	Year	Country	Document Type	Empirical Data or Findings	Suicide Prevention Recommendations
Chau [32]	2020	Australia	Submission to government on suicide prevention	None	Compulsory for education providers to provide health services to international students Provide access to mental health services to international students Educate GPS to refer international students to appropriate mental health services Education for international students on mental health Mental health services to provide culturally appropriate services
Council of International Students Australia [33]	2019	Australia	Government Submission	None	Implement specific reporting framework/protocol for institutions in dealing with international students’ suicides to better track scope of problem
English Australia [34]	2018	Australia	Best practice guide	None	Staff training in mental health issues and risk factors for suicide in international students Colleges require clear protocols for crisis situations
Haas et al. [35]	2005	USA	Evidence based guide to Suicide prevention	None	Screening is an important strategy and should be targeted at risk groups, such as international students Promote sense of belonging for international students who may experience isolation through collaborative efforts between college staff across all organizational levels
Jamieson [36]	2019	Australia	Coroner’s Report	Design: Case Series Study Aim: To assess international student suicides between 2009 and 2015 in Australia Participants: 27 international students who died by suicide, with 24 from Asia, 2 from Americas, and 1 from Europe Results: Males aged 18–24 years (15 of 27), followed by males aged 25–29 years (5 of 27), were most common group. International students who died by suicide were less likely to have known history of self-harm, a mental health diagnosis, accessed services for mental health prior to suicide, or had a previously documented suicide attempt than domestic students who died by suicide. International students were more likely to experience educational or financial stressors, but less likely to experience death of a family member, conflict with a family member, exposure to family violence, or conflict with non-family acquaintances than domestic students before suicide. Most common stressor for international students before suicide was educational, with course failure and fear of telling parents particularly common. Similar proportions of students in the international and Australian-born cohorts gave indicators of intent before suicide; experienced interpersonal stressors such as separation from partner and conflict with partner prior to death; and experienced work-related stressors, social isolation, and substance misuse	Governments and education departments should work with relevant stakeholders to identify strategies to engage international students with mental health support Governments and education departments should work with relevant stakeholders to identify strategies to engage international students with mental health support Universities should provide reports of incidents within 4 weeks
Jamieson [11]	2021	Australia	Coroner’s Report	Design: Case Series Study Aim: To assess international student suicides between 2009 and 2019 in Australia Participants: 47 international students who died by suicide with 37 from Asia, 4 from Africa, 4 from the Americas, and 2 from Europe Results: Majority of suicides were male (70.2%) and under 24 years of age (63.8%). Most common stressors were educational (63.8%; e.g., course failure, course direction) and financial (32.6%). Intersecting stressors including homesickness and social isolation (22.4%) and parental expectations (14.3%) were commonly related to course failure and academic stress	Collaborations between university supports and external services including formal treatment pathway development Community engagement and linking as first point of support Training for students and providers in mental health access and barriers–potentially facilitated by insurers Counseling services to meet the language and cultural diversity needs of their international students
Koo [37]	2010	USA	Master’s thesis	None	Enhance counseling staff awareness of cultural attitudes to help seeking and mental health problems, and their own potential biases Create supportive and inclusive atmosphere at counseling centers to help students form connections in host country Counseling centers should take proactive approach to international students through organizing activities, workshops, and support groups
Lento [38]	2016	USA	Doctoral thesis	Design: Cohort study Aim: To investigate the role of ambivalence in the suicidal mind and its usefulness in stratifying risk among suicidal college students Participants: 226 university students (17% international students) attending university counseling center. M_age_ = 21.42, SD = 3.82. 44.2% male, 54.9% female, 0.8% other. Country of origin of international students not specified Results: International students more likely to withdraw from counseling or university prior to resolution of suicidal ideation than domestic students	Cultural competence training for mental health professionals to maintain international student engagement with services
Maramaldi [39]	2019	USA	Book chapter	None	Staff training to recognize suicide warning signs in international students Staff training on suicide risk assessment Encourage identification and reporting of suicide warning signs in students and staff Educational organizations should have a suicide protocol for if and when suicide occurs
McCauley [40]	2022	USA	Report of a review of suicide prevention protocols at educational institute	None	Suicide risk screening should be part of counseling intake process and provided in students’ native languages where possible Suicide prevention training should be provided to all incoming students, including international students
Orygen [41]	2020	Australia	Report	Collected data on international student mental health and physical safety but not specific to suicide	Provide gatekeeper training (e.g., Mental Health First Aid) to support staff and volunteers who work with vulnerable international students
Poole and Robinson [42]	2019	UK	Non-peer reviewed journal article—opinion piece	None	Student societies can be used to identify barriers and encourage help seeking in international student groups
Public Health England [43]	2019	UK	Practice resource	None	Postvention for clusters should include different relevant cultural groups. Recruitment may occur through international office
Substance Abuse and Mental Health Services Administration (SAMHSA) [44]	2021	USA	Evidence-based resource guide for universities	None	Focused outreach efforts to increase knowledge of available mental health services should be targeted at international students Providers should refer non-native English speakers to services in their own language Where possible, mental health service providers should have same cultural background as student and interpreters can be used in the case of a language barrier
Suicide Prevention Australia [45]	2021	Australia	Submission to government on suicide prevention	None	Collaboration between governments, international student bodies, and local agencies should be used to roll out programs for international students
Suicide Prevention Australia [46]	2021	Australia	Policy position statement	None	Collaboration between government and organizations with expertise (e.g., lived experience, carers, and persons involved in family and international student support) in culturally appropriate service delivery to design, implement, and evaluate services
Suicide Prevention Resource Center [47]	2004	USA	White paper	None	Cultural competence training should be provided to mental health staff that identifies risks for international student mental health and suicide, including language issues, financial pressures, and culturally related mental health stigma
The JED Foundation [48]	2006	USA	Framework for developing institutional protocols for distress/suicide	None	Include international office in suicide prevention protocol development, implementation, use, and review Consider assessment barriers such as language and cultural differences Include translator where necessary to engage relevant parties
Xiong [49]	2018	USA	Doctoral thesis	Design: Case control study Aim: To investigate the mental health of Asian international students in the U.S. through a nationwide sample Sample: 10,731 university students with Asian international (n = 3702; 48% male, 51.2% female, 0.4% other, 0.4% missing), American (n = 3649; 48.4% male, 51.3% female, 0.4% other, 0.4% missing), and other international (n = 3380; 44.3% male, 55.4% female, 0.3% other) students. Country of origin not reported for international students Results: Proportion of Asian international students reporting self-injury, considering suicide, and dying by suicide was higher than in domestic and other international student groups. Asian international students sought less mental health services and were less willing to seek those services than American students and other international students	Mental health professionals and faculty members should be trained in predictors of poor mental health in international students Train staff who have regular contact with international students in their in mental health needs (e.g., residential assistants) Collaboration between student organizations and university services should be used to produce effective outreach programs or workshops that reduce mental health stigma

**Table 3 ijerph-20-01500-t003:** Summary of International Student Suicide Prevention Recommendations.

Cultural Competency Training on Suicide and Provision of Culturally Sensitive Services	Improved and Increased Risk Screening for Suicide	Proactive Intervention and Engagement Strategies	Collaborative Approaches to Streamline Service Access and Improve Available Support
Target groups for training: - Mental health practitioners/counsellors [22,27,30,37,38,47,49] - Academics [23,49] - Course advisors [23,27,49] - Other staff who work with international students (e.g., student residence staff, student leaders) [23,34,41,49] - General Practitioners (GPs) [22,32] Key training topics: - Signs of risk for suicide in international students [27,31,34,39,47] - Risk factors for suicide in international students (racism/ethnic discrimination, isolation/loneliness, campus belongingness, sense of burden, unmet interpersonal needs, maladaptive perfectionism academic stressors, cross-cultural loss, and financial issues) [12,23,27,28,29,47] - Barriers to help seeking and accessing support (e.g., mental health stigma, attitudes towards help seeking) [11,37,42,47,48] Culturally sensitive service provision: - Culturally sensitive mental health services should be offered by education providers. This includes creating supportive and inclusive counselling service environments that increase international student engagement and where possible providing mental health service providers with same cultural background as students [11,37,44]	Increased use of formal suicide risk screening measures: - At mental health center intake [1,40] Increase risk screening in university community by: - Mental health practitioners [40] - University staff [39] - International students and their communities [41,49] Development of new or culturally relevant risk measures: - Risk screening measures in multiple languages [40] - Risk screening measures that include key risk factors for suicide in international students [22]	Targeted interventions: - Orientation programs that include information regarding mental health services and guidance on the broader local health system [44] - Information campaigns for international students targeting cultural misconceptions of mental health, help seeking, and challenges with acculturation [1,8,10,25,29,32] - Campaigns targeting other risk factors for suicide such as sense of belongingness, loneliness or burdensomeness [10,12,25,30,35,37] - Campaigns targeting protective factors for suicide such as problem focused coping [25] - Multilanguage psychoeducation materials [11,24,31] - Increased resources to support campaigns that increase international student engagement with services [21] Peer based and other collaborative outreach programs: - Peer based mental health awareness and referral training to reduce stigma and increase service engagement [31,40] - Link students with existing student and community networks to enhance belonging and social connectedness (e.g., buddy programs, international student networks) [11,12,25,28,37,41]	Collaboration between key services and groups with expertise: - Collaboration between governments, international student bodies, local agencies (e.g., counselling services, international student office, academic affairs, diversity office), and other expert groups (e.g., international students with lived experience, carers, and persons involved in international student support) should be used to develop, implement, and evaluate suicide prevention programs and services for international students [12,24,28,31,35,36,45,46,49] Streamlined mental health pathways for international students: - Coordinated culturally sensitive response between universities, community mental health services, and emergency departments for international students [11] - Identify and address key barriers to service access such as cost and insurance coverage for mental health services through insurers [11]

Notes: The experts who reviewed these recommendations suggested further potential areas for suicide prevention that were not identified in the literature for this review. These recommendations were: (1) utilizing institutional social media channels to communicate help seeking and other mental health messaging relevant to international students; (2) providing pre-departure mental health and service information to international students before arrival in their destination country.

## Data Availability

Not applicable.

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
