# Peer review of "Suicide Prevention for International Students: A Scoping Review"

_ijerph, 2023, doi:10.3390/ijerph20021500_

Round 1
Reviewer 1 Report
This paper is written to a very high standard in an important area of practice. Having word in the higher education sector with international students for nearly 20 years, there are very few examples of literature reviews that draw together key learnings and findings from existing studies. As such, this is a welcome addition to the field.
The writing is accessible and engaging - not always easy to achieve in such a paper - and the review is well-contextualised. The methods are appropriate and clearly outlined and the results presented in a way consistent with the method. The papers are well-summarised and discussion and conclusions reached consistent with the results.
I would recommend acceptance of the paper in its current form. One minor addition might be to strengthen the assertion as to the limited efficacy of risk assessment tools and suicide prediction. Indeed, in the UK NICE's latest guidance, (September 2022) drawing on the literature re such tools takes a position that actively advises against the use of such tools in suicide prediction, prevention, allocation of services and also differentiating between low, medium and high risk. As such, this will be a notable change in mental health practice, which will also have implications in education settings and thus, with international students. While not directly relevant to the aims and scope of this paper, it might help strengthen that particular aspect of the discussion.
Beyond that minor suggestion, an important paper that makes a helpful contribution to academia and practice.
Author Response
Reviewer 1 Comments |
Author Responses |
This paper is written to a very high standard in an important area of practice. Having word in the higher education sector with international students for nearly 20 years, there are very few examples of literature reviews that draw together key learnings and findings from existing studies. As such, this is a welcome addition to the field. The writing is accessible and engaging - not always easy to achieve in such a paper - and the review is well-contextualised. The methods are appropriate and clearly outlined and the results presented in a way consistent with the method. The papers are well-summarised and discussion and conclusions reached consistent with the results.
|
We thank the reviewer for their positive comments and feedback. |
One minor addition might be to strengthen the assertion as to the limited efficacy of risk assessment tools and suicide prediction. Indeed, in the UK NICE's latest guidance, (September 2022) drawing on the literature re such tools takes a position that actively advises against the use of such tools in suicide prediction, prevention, allocation of services and also differentiating between low, medium and high risk. As such, this will be a notable change in mental health practice, which will also have implications in education settings and thus, with international students. While not directly relevant to the aims and scope of this paper, it might help strengthen that particular aspect of the discussion. |
We have added the following sentence to the discuss in the relevant section: |
Reviewer 2 Report
This is an interesting article. I have a few minor comments.
1. I'd note that sample size is relatively small for relevant studies in the literature.
2. I think it may be important to mention that the selection of international students differs legally based on each country. Do you think there is mental health selection of international students, that is international students are healthier than domestic students at the time of the arrival? Discussing this in introduction may strengthen the study.
Author Response
Reviewer 2 Comments |
Author Responses |
1. I'd note that sample size is relatively small for relevant studies in the literature. |
Thank you for your feedback. We have added the following to the 4.1.1 Characteristics of Existing Evidence section:
“Study sample sizes in many studies were also small, reducing power to identify small significant effects or generalize findings.” |
2. I think it may be important to mention that the selection of international students differs legally based on each country. Do you think there is mental health selection of international students, that is international students are healthier than domestic students at the time of the arrival? Discussing this in introduction may strengthen the study. |
To the best of our knowledge, mental health screening is not a key component of the selection process for international students in most countries. While health insurers may request such information, it is not available publicly. The available evidence comparing mental health of domestic and international students is somewhat mixed, with some evidence suggesting that international students have better mental health than their domestic counterparts, while other evidence suggesting the opposite. It is also very complex to draw strong conclusions about this evidence because international students are more likely to stigmatise mental health issues and the available measures are not necessarily well adapted to migrant communities, as noted in the current review findings. Given the mixed evidence, and lack of available information on mental health in student selection, we have not added any information regarding this to the introduction. However, we will take these ideas into consideration in our future work. |
Reviewer 3 Report
Thank you to the journal for the opportunity to review this article. Before I even begin reading it, I think that this was already a very important topic, but made much more important in the context of the pandemic. International students may not be reported in headline suicide figures as they are not permanent residents. Denominators for this group may be hard to come by, so, increases or decreases in suicide numbers or rates may go unnoticed without careful attention. Then, they have the added pressure of sometimes needing to send money home, their family’s hopes and expectations of their success, perhaps learning a new language, adapting to life and the demands of study in a new culture, and the degrees from their original countries not necessarily being recognised in the countries they study in. The countries they come from may of course have less mental health literacy or less developed mental health care systems, all contributing to a heightened risk, especially when the pandemic occurred, which cut out their study, support systems, ability to see family and potentially their ability to earn money. It is in that sense that this article is extremely timely and important. I know the authors know all this.
Page 1, line 11: ‘systematic scoping review’ – per the seminal article on types of reviews, there was no systematic scoping review at the time: Grant, M.J. and Booth, A. (2009), A typology of reviews: an analysis of 14 review types and associated methodologies. Health Information & Libraries Journal, 26: 91-108.
Maybe this is a newer development that the two have been paired, and I guess the authors will elaborate on which elements were systematic.
Page 1, lines 17 to 19: ‘No studies’ - This is such a disappointing finding for you (and everyone), isn’t it! When first reading it, it makes me wonder if we need to shift to looking at the composition of international students, in terms of country of birth and ethnic background, and see if there are any individual suicide prevention interventions or reviews of them for migrants from different countries, noting that findings may not translate exactly as the pressures of a student are distinct from a migrant worker (nevertheless, until we have an evidence base, this may be our best bet).
Page 1, line 35: Just may need to slightly clarify who you mean by ‘some groups’. Do you mean some groups in higher education/university, or some groups of international students? I presume the former.
Page 2, line 65: Do you need a comma after ‘collating’, for readability?
Page 2, lines 82 to 83: did you search via EBSCOhost? Minor detail.
Page 2, line 90: Probably just need to spell out OSF for uninitiated readers.
Page 2, line 91: You mentioned ProQuest Dissertations & Theses Global on line 83, so maybe you don’t need to mention it again here?
Page 2, line 92 to 93: Can you clarify here if you put these keywords in separately and combined with the ‘AND’ operator, or you just listed these in the database as individual words or one phrase? I understand some of the databases would not enable ‘AND’ and ‘OR’ operators, so you may be limited here.
Page 3, line 110: There is a suggested citation for Covidence: https://support.covidence.org/help/how-can-i-cite-covidence
Page 3, line 132: It reads as though there were five stakeholders + two provider partners + three subject matter experts for a total of 10, but I think you mean 2 + 3 = 5? I think you can omit ‘A total of five stakeholders,’. It’d be good to say who the two provider partners and three subject matter experts were if they consent to that. Do you mean provider of tertiary education, or non-clinical support services for international students, or clinical support services for them?
Page 3, line 138: Given the limited studies retrieved, the duplicate removal process has additional importance. To clarify who did it you may want to rephrase ‘duplicates were removed in Covidence’ to ‘Covidence removed duplicates’ and clarify if you double-checked to confirm it had correctly identified duplicates, as Covidence has imperfect sensitivity: https://systematicreviewsjournal.biomedcentral.com/articles/10.1186/s13643-021-01583-y/McKeown, S., Mir, Z.M. Considerations for conducting systematic reviews: evaluating the performance of different methods for de-duplicating references. Syst Rev 10, 38 (2021). https://doi.org/10.1186/s13643-021-01583-y
Page 3, line 138: It says 30 studies here, but 31 in the flowchart? It also says in the flowchart you couldn’t locate two, so 29 full-text records were screened rather than 30?
Page 4, line 143: Since you’ve retrieved way more records through the grey literature search, how these records were managed is crucial. I therefore think you need to revise the flowchart so it looks more like Fig. 1 in the PRISMA 2020 Statement: https://systematicreviewsjournal.biomedcentral.com/articles/10.1186/s13643-021-01626-4/figures/1. You might be able to do this in the SHINY app developed for this: https://estech.shinyapps.io/prisma_flowdiagram/. See , 2022). PRISMA2020: An R package and Shiny app for producing PRISMA 2020-compliant flow diagrams, with interactivity for optimised digital transparency and Open Synthesis. Campbell Systematic Reviews, 18, e1230. , , & (
I did try clicking on the supplementary materials link, but it takes me to a 404 page: www.mdpi.com/xxx/s1 Error 404 - File not found
The webpage you are looking for could not be found. The URL may have been incorrectly typed, or the page may have been moved into another part of the mdpi.com site.
Return to the main page
Contact
Page 5, line 149: In Table 1, I think you can omit the bullet points as the indentation loses a lot of room for you and aligning bullet points in tables is notoriously difficult. Just use spaces for separate points?
Page 6, bottom of table, Kaslow et al. 2012: the bottom of column 4 says “development effective”
Page 8 and 9 – Table: I note that there is perceived burdensomeness and thwarted belongingness in some studies. I therefore wonder if any theories of suicidal behaviour could be discussed in the discussion section.
Page 18, line 198: Just check grammar here: A key component of this approach utilisation of peer-based programs.
Page 19, line 205: ‘suicide protocol’ – just to clarify, is this the protocol for when a suicide occurs, to reduce contagion?
Page 23, line 353: Could you actually apply the Cochrane Risk of Bias 2 tool to these studies, since none of them are interventional? If no randomisation has occurred, any intervention study might start out having a high risk of selection bias without the need to apply that tool? But because you don’t have any interventions, I think you could omit this point.
Page 22, lines 347 to 349: In a subsequent paper, it’d be interesting for you to create a dataset of these recommendations, coding which organisation they came from, the backgrounds of the people or institutions who made them, what the recommendation is based on (any literature?), and tabulate and visualise the recommendations. If this scoping review is your reconnaissance mission, and about description, don’t let the thorough search go to waste! I had a look at the description of a critical review and you might be able to do one of those with the corpus you have. A typology of reviews: an analysis of 14 review types and associated methodologies
Maria J. Grant, Andrew Booth
Because there are no interventions yet, it’s crucial we have academics with expertise in suicide prevention critically review this literature (in terms of who made what recommendations and what the consensus is – what are the most common recommendations) as well as describe it.
Lastly, I got the sense (maybe mistakenly) that this review focused on implications for support services at universities. I wonder what learnings or implications there are from state government transcultural mental health centres, existing societies for international students from specific countries, and multicultural not-for-profits that don’t exclusively serve asylum seekers and refugees. To me, these people have the real expertise of what it’s like to be a migrant in the countries the international students study in. It may be hard for universities to provide culturally-specific or culture-based interventions for international students given the varied countries they come from. Even if they can, surely there is much to be gained by drawing on the expertise of the government and the existing community of people from that country in the host country. This is recognising that suicide prevention is everyone’s business and that working together, hopefully international students won’t fall through the cracks.
Merry Christmas and happy new year to the authors – I hope they have a nice and restful break. I hope I get an opportunity to reread this manuscript.
Author Response
Reviewer 3 |
Author Responses |
Thank you to the journal for the opportunity to review this article. Before I even begin reading it, I think that this was already a very important topic, but made much more important in the context of the pandemic. International students may not be reported in headline suicide figures as they are not permanent residents. Denominators for this group may be hard to come by, so, increases or decreases in suicide numbers or rates may go unnoticed without careful attention. Then, they have the added pressure of sometimes needing to send money home, their family’s hopes and expectations of their success, perhaps learning a new language, adapting to life and the demands of study in a new culture, and the degrees from their original countries not necessarily being recognised in the countries they study in. The countries they come from may of course have less mental health literacy or less developed mental health care systems, all contributing to a heightened risk, especially when the pandemic occurred, which cut out their study, support systems, ability to see family and potentially their ability to earn money. It is in that sense that this article is extremely timely and important. I know the authors know all this. |
We thank the reviewer for highlighting the importance of this topic. |
Page 1, line 11: ‘systematic scoping review’ – per the seminal article on types of reviews, there was no systematic scoping review at the time: Grant, M.J. and Booth, A. (2009), A typology of reviews: an analysis of 14 review types and associated methodologies. Health Information & Libraries Journal, 26: 91-108.
Maybe this is a newer development that the two have been paired, and I guess the authors will elaborate on which elements were systematic. |
We have removed the term “systematic” from the abstract when describing the review. |
Page 1, lines 17 to 19: ‘No studies’ - This is such a disappointing finding for you (and everyone), isn’t it! When first reading it, it makes me wonder if we need to shift to looking at the composition of international students, in terms of country of birth and ethnic background, and see if there are any individual suicide prevention interventions or reviews of them for migrants from different countries, noting that findings may not translate exactly as the pressures of a student are distinct from a migrant worker (nevertheless, until we have an evidence base, this may be our best bet). |
This is a good point. Encouragingly, the recommendations for international students align with findings from a recent systematic review on suicide prevention with CALD communities and we added a point about this in the discussion: “There is however an extensive range of recommendations aimed at reducing international student suicide available in both the academic and grey literature that align with the broader literature on suicide prevention with individuals from culturally and linguistically diverse (CALD) communities.”
|
Page 1, line 35: Just may need to slightly clarify who you mean by ‘some groups’. Do you mean some groups in higher education/university, or some groups of international students? I presume the former. |
We have amended this sentence to read: “However, some groups of students in higher education settings may require specific suicide prevention programs that are designed to meet their unique needs.” |
Page 2, line 65: Do you need a comma after ‘collating’, for readability? |
Yes, we have added the missing comma. |
Page 2, lines 82 to 83: did you search via EBSCOhost? Minor detail. |
We have revised this section to note Ebscohost rather than Ebsco for the relevant databases. |
Page 2, line 90: Probably just need to spell out OSF for uninitiated readers. |
We have included the full term “Open Science Framework” in this section now. |
Page 2, line 91: You mentioned ProQuest Dissertations & Theses Global on line 83, so maybe you don’t need to mention it again here? |
We have removed the full database title on line 91, as suggested. |
Page 2, line 92 to 93: Can you clarify here if you put these keywords in separately and combined with the ‘AND’ operator, or you just listed these in the database as individual words or one phrase? I understand some of the databases would not enable ‘AND’ and ‘OR’ operators, so you may be limited here. |
The two search phrases listed in the manuscript (e.g., “international student suicide” and “international student suicide prevention”) were entered into each database as separate searches. No ‘AND’ or ‘OR’ operators were used. We did this to have a consistent strategy, as many of the databases did not allow for the use of ‘AND’ or ‘OR’ operators. To make this clearer, we have now revised this sentence to read “The two sets of keywords used for all grey literature searches were “international student suicide” and “international student suicide prevention”” |
Page 3, line 110: There is a suggested citation for Covidence: https://support.covidence.org/help/how-can-i-cite-covidence |
We have added the Covidence reference as per the reviewer’s suggestion. |
Page 3, line 132: It reads as though there were five stakeholders + two provider partners + three subject matter experts for a total of 10, but I think you mean 2 + 3 = 5? I think you can omit ‘A total of five stakeholders,’. It’d be good to say who the two provider partners and three subject matter experts were if they consent to that. Do you mean provider of tertiary education, or non-clinical support services for international students, or clinical support services for them? |
We have amended this sentence as per the suggestion and it now reads: “Two provider partners and three subject matter experts reviewed and provided feedback on the results of the review to ensure that our interpretations were applicable, understandable, and included all relevant documents and information.” |
Page 3, line 138: Given the limited studies retrieved, the duplicate removal process has additional importance. To clarify who did it you may want to rephrase ‘duplicates were removed in Covidence’ to ‘Covidence removed duplicates’ and clarify if you double-checked to confirm it had correctly identified duplicates, as Covidence has imperfect sensitivity: https://systematicreviewsjournal.biomedcentral.com/articles/10.1186/s13643-021-01583-y/McKeown, S., Mir, Z.M. Considerations for conducting systematic reviews: evaluating the performance of different methods for de-duplicating references. Syst Rev 10, 38 (2021). https://doi.org/10.1186/s13643-021-01583-y |
Thank you for pointing this out. The duplicate removal process included both automatic (e.g., Covidence’s tool) and manual duplicate removal. All records that were identified as potential duplicates automatically by Covidence were manually checked. We have revised this section to note this process more clearly: “There were 74 records retrieved through our search of the peer reviewed literature and 14 duplicates were removed using Covidence's duplicate removal function after all potential duplicates identified by the system were manually reviewed.”. |
Page 3, line 138: It says 30 studies here, but 31 in the flowchart? It also says in the flowchart you couldn’t locate two, so 29 full-text records were screened rather than 30? |
Thank you for identifying this error, we have updated the text to correct number. |
Page 4, line 143: Since you’ve retrieved way more records through the grey literature search, how these records were managed is crucial. I therefore think you need to revise the flowchart so it looks more like Fig. 1 in the PRISMA 2020 Statement: https://systematicreviewsjournal.biomedcentral.com/articles/10.1186/s13643-021-01626-4/figures/1. You might be able to do this in the SHINY app developed for this: https://estech.shinyapps.io/prisma_flowdiagram/. See Haddaway, N. R., Page, M. J., Pritchard, C. C., & McGuinness, L. A. (2022). PRISMA2020: An R package and Shiny app for producing PRISMA 2020-compliant flow diagrams, with interactivity for optimised digital transparency and Open Synthesis. Campbell Systematic Reviews, 18, e1230. |
We have updated the figure to the suggested version. |
I did try clicking on the supplementary materials link, but it takes me to a 404 page: www.mdpi.com/xxx/s1 Error 404 - File not found
The webpage you are looking for could not be found. The URL may have been incorrectly typed, or the page may have been moved into another part of the mdpi.com site.
Return to the main page Contact |
Thanks for noting this issue. We have removed this section from the revised manuscript, as there are no supplementary files. |
Page 5, line 149: In Table 1, I think you can omit the bullet points as the indentation loses a lot of room for you and aligning bullet points in tables is notoriously difficult. Just use spaces for separate points? |
Thanks for this suggestion. We removed bullet points in tables 1 and 2. We also tried replacing the bullet points with spaces in table 3, but the table was then spread across multiple pages making it more difficult to follow. For this reason, we have reworked table 3 to use bullet points in a way that is clearer and takes less space than before. |
Page 6, bottom of table, Kaslow et al. 2012: the bottom of column 4 says “development effective” |
We have revised this sentence to include the missing words. It now reads: “University staff and coalition members (unspecified number & no demographic information provided) who participated in the development of an effective campus wide coalition for suicide prevention” |
Page 8 and 9 – Table: I note that there is perceived burdensomeness and thwarted belongingness in some studies. I therefore wonder if any theories of suicidal behaviour could be discussed in the discussion section. |
We have updated to the discussion to note the current use of the interpersonal theory of suicide in the literature:
“Finally, some studies drew upon existing theories of suicide, with most of these utilising Joiner’s interpersonal theory of suicide and the key associated variables of perceived burdensomeness and belonging”
We have also added a paragraph with recommendations regarding the use of a broader range of theories in future research:
Opportunities also exist to draw upon a wider range of theories and models from the suicide, acculturation, and public health literature when developing and testing potential interventions or prevention programs for international students. For example, comprehensive suicide prevention strategies often draw upon a public health framework that stratifies prevention into universal (i.e., delivered to the whole population), selective (i.e., delivered to groups at higher risk of suicide), and indicated (i.e., delivered to people showing suicide related behaviours) approaches [60]. Combined approaches addressing each of the levels of this framework have been shown to be effective with young people [60], and this model could be used to develop comprehensive suicide prevention interventions for international students. |
Page 18, line 198: Just check grammar here: A key component of this approach utilisation of peer-based programs. |
We have amended this sentence, it now reads: “A key component of this approach is the utilisation of peer-based programs.” |
Page 19, line 205: ‘suicide protocol’ – just to clarify, is this the protocol for when a suicide occurs, to reduce contagion? |
We have amended this to now read: “Educational organisations should have a suicide protocol for if and when suicide occurs”. |
Page 23, line 353: Could you actually apply the Cochrane Risk of Bias 2 tool to these studies, since none of them are interventional? If no randomisation has occurred, any intervention study might start out having a high risk of selection bias without the need to apply that tool? But because you don’t have any interventions, I think you could omit this point. |
We have removed this section, as per your suggestion. |
Page 22, lines 347 to 349: In a subsequent paper, it’d be interesting for you to create a dataset of these recommendations, coding which organisation they came from, the backgrounds of the people or institutions who made them, what the recommendation is based on (any literature?), and tabulate and visualise the recommendations. If this scoping review is your reconnaissance mission, and about description, don’t let the thorough search go to waste! I had a look at the description of a critical review and you might be able to do one of those with the corpus you have. A typology of reviews: an analysis of 14 review types and associated methodologies
Maria J. Grant, Andrew Booth
Because there are no interventions yet, it’s crucial we have academics with expertise in suicide prevention critically review this literature (in terms of who made what recommendations and what the consensus is – what are the most common recommendations) as well as describe it. |
Thank you for this suggestion. We will look into ways that we work with this data further in the future. |
Lastly, I got the sense (maybe mistakenly) that this review focused on implications for support services at universities. I wonder what learnings or implications there are from state government transcultural mental health centres, existing societies for international students from specific countries, and multicultural not-for-profits that don’t exclusively serve asylum seekers and refugees. To me, these people have the real expertise of what it’s like to be a migrant in the countries the international students study in. It may be hard for universities to provide culturally-specific or culture-based interventions for international students given the varied countries they come from. Even if they can, surely there is much to be gained by drawing on the expertise of the government and the existing community of people from that country in the host country. This is recognising that suicide prevention is everyone’s business and that working together, hopefully international students won’t fall through the cracks. |
We have revised research and policy directions section of the discussion to better highlight the broader implications for government, mental health service providers and other organisations who work with international students. |